# Design and Evaluation of Dissolvable Microneedles for Treating Atopic Dermatitis

**DOI:** 10.3390/pharmaceutics15041109

**Published:** 2023-03-31

**Authors:** Noa Ben David, Yuval Richtman, Adi Gross, Ruba Ibrahim, Abraham Nyska, Yuval Ramot, Boaz Mizrahi

**Affiliations:** 1Faculty of Biotechnology and Food Engineering, Technion-Israel Institute of Technology, Haifa 3200003, Israel; 2Department of Dermatology, Hadassah Medical Center, Jerusalem 9112001, Israel; 3Faculty of Medicine, Hebrew University of Jerusalem, Jerusalem 9112001, Israel; 4Sackler School of Medicine, Tel Aviv University, Tel Aviv 6200515, Israel

**Keywords:** dissolving microneedles, atopic dermatitis, drug release, dexamethasone, PVA, PVP

## Abstract

Atopic dermatitis (AD) is a chronic inflammatory skin disease caused predominantly by immune dysregulation. The global impact of AD continues to increase, making it not only a significant public health issue but also a risk factor for progression into other allergic phenotype disorders. Treatment of moderate-to-severe symptomatic AD involves general skin care, restoration of the skin barrier function, and local anti-inflammatory drug combinations, and may also require systemic therapy, which is often associated with severe adverse effects and is occasionally unsuitable for long-term use. The main objective of this study was to develop a new delivery system for AD treatment based on dissolvable microneedles containing dexamethasone incorporated in a dissolvable polyvinyl alcohol/polyvinylpyrrolidone matrix. SEM imaging of the microneedles showed well-structured arrays comprising pyramidal needles, fast drug release in vitro in Franz diffusion cells, an appropriate mechanical strength recorded with a texture analyzer, and low cytotoxicity. Significant clinical improvements, including in the dermatitis score, spleen weights, and clinical scores, were observed in an AD in vivo model using BALB/c nude mice. Taken together, our results support the hypothesis that microneedle devices loaded with dexamethasone have great potential as a treatment for AD and possibly for other skin conditions as well.

## 1. Introduction

Atopic dermatitis (AD) is a chronic inflammatory skin disease characterized by skin barrier dysfunction, erythema, edema formation, dryness, and pruritus [1,2] and associated with infiltration of inflammatory mediators such as eosinophils, neutrophils, and mast cells [3]. AD, which affects up to 20% of children and 7% of adults, places a substantial physical, emotional, and financial burden on patients and their families [4]. Treatment of AD aims to reduce the frequency, duration, and severity of symptoms, and is applied in a step-wise manner [5]. Patients with a genetic tendency to develop AD are advised on basic skin care regimens, including liberal and frequent use of moisturizers [6]. Treating moderate-to-severe AD involves restoration of the skin barrier function and drug combinations that suppress local inflammation and prevent local or systemic superinfection [7]. Although topical corticosteroids are considered as the first line of pharmacological treatment, there is a subgroup of patients whose AD does not improve despite increasing applications of topical steroids [8]. In moderate-to-severe disease, systemic immunosuppressants, with or without systemic steroids, are often prescribed. These treatments may be beneficial for many of the patients; however, they might be associated with severe adverse effects, require careful monitoring, and some of them are not suitable for long-term use (e.g., cyclosporine) [7]. More effective and well-tolerated therapies for AD are therefore required [9].

Microneedles (MNs) are minimally invasive microscopic applicators used as a painless alternative to hypodermic needles [10]. They are made of a variety of materials and vary in size, shape, morphology, and functionality [11,12]. Since dissolvable MNs can painlessly bypass the skin’s epidermis, dissolve at the treated area, and release drugs directly to the inner skin layers [13,14,15], they are often used as common drug delivery systems [16]. Polyvinyl alcohol (PVA) and polyvinylpyrrolidone (PVP) have been chosen as the matrix thanks to their water soluble and hygroscopic nature, making them ideal for a wide range of skin care and cosmetic products in which it is crucial to maintain moisture [17,18,19]. Both PVA and PVP are considered ideal for fabricating fibers, films, and MNs due to their semi-crystalline structure and excellent physical properties [20,21,22]. In addition, the hygroscopic properties of the polymers provide a moisturizing action when used in dermal applications [19]. MNs fabricated from PVA might have insufficient mechanical properties to penetrate the stratum corneum. In order to increase the mechanical strength of the MNs, addition of another polymer (e.g., PVP) to the MN composition is often efficient. MNs made from a mixture of PVA and PVP present improved mechanical properties since a hydrogen bond can be formed between the carbonyl group of PVP and the hydroxyl group of PVA [23]. In addition, PVA and PVP are biocompatible with skin cells and FDA approved for pharmaceutical and biomedical applications [24,25] with no toxicity to skin cells [26,27].

We hypothesized that MNs made from dissolvable polymers, that release steroids into and on top of the skin, may overcome some of the shortcomings of currently used formulations. The abilities of the hygroscopic materials, which are capable of preserving humidity and form a protecting barrier on the skin, may further contribute to the healing process. We hereby present dissolvable MNs made from mixtures of low molecular weight PVA and high molecular weight PVP and loaded with the corticosteroid dexamethasone (Dex). We first describe the fabrication and formula optimization in terms of morphology and mechanical properties, and then report the drug release pattern and the ability of the MNs to fix onto a skin tissue while releasing the drug in situ. Finally, to evaluate the potential utilization of this system for treating AD, we describe in vitro and in vivo studies in which we compare the performance of the Dex-MNs with the empty MN and non-treated groups using an AD mouse model.

## 2. Materials and Methods

### 2.1. Materials

PVA (Mw = 13,000–23,000 g·mol^−1^, 99% hydrolysis), PVP, K90 (Mw = 360,000 g·mol^−1^), 2,4-dinitrochlorobenzene (DNCB), gelatin from porcine skin, Dulbecco’s phosphate-buffered saline (PBS), Dulbecco’s modified Eagle’s medium (DMEM), fetal bovine serum, sodium acetate trihydrate, rhodamine B, acetone, formalin haematoxylin and eosin (H&E), and xylene were purchased from Sigma Aldrich (Sigma Chemicals, St. Louis, MO, USA). Other materials (and their sources) were CellTiter 96 solution kit (Promega, Fitchburg, WI, USA), olive oil (Zeita, Israel), dexamethasone-21-phosphate disodium salt (Dex) (Alfa Aesar, Ward Hill, MA, USA), acetonitrile (JT Baker, Phillipsburg, NJ, USA), and optical cutting temperature (OCT) compound (Scigen, Gardena, CA, USA).

### 2.2. Fabrication and Characterization

#### 2.2.1. Fabrication

PVA/PVP MNs were prepared using the micromolding technique [28]. In brief, various aqueous solutions of fixed 10 %wt were prepared with different PVA/PVP ratios (e.g., 1:2 and 1:1) and only PVA or PVP. PVA/PVP solutions with a 1:2 ratio were mixed with dexamethasone to a final concentration of 0.25% and samples were vortexed. Then, solutions (110 µL) were casted on an 8 mm × 8 mm PDMS mold (Micropoint Technologies Pte Ltd., Singapore) with a cavity height of 500 μm and a base diameter of 200 μm. Molds were spun (Laurell model WS-650MZ-23NPPB, North Wales, PA, USA) at 4000 rpm for 1 min, followed by recasting (110 µL) and a second spin cycle. Molds were filled with solutions (110 µL) and were allowed to dry for 24 h at room temperature followed by careful removal of the MNs using tweezers. MNs were stored in a desiccator until use. Control MNs of the same composition but without dexamethasone were fabricated and stored under similar conditions.

#### 2.2.2. Characterization

The morphology of the MNs was characterized using a wide field scanning electron microscope (FEI E-SEM Quanta 200, Eindhoven, The Netherlands) and a high-resolution scanning electron microscope (Zeiss Ultra-Plus FEG-SEM, Tescan, Brno, Czech Republic). The mechanical behavior of each of the compositions during compression and tension tests [29] was evaluated using a Lloyd TA1 texture analyzer (Lloyd Instruments Ltd., West Sussex, UK) equipped with a 10 N load cell for compression and a 2500 N load cell for the tension test. Microneedles were fixed at the center of the bottom plate (11.5 cm diameter), with needles pointing up, using a carbon patch. The upper plate began compressing the MNs at a speed of 1 mm/min (n = 4) up to 330 μm from the contact position. The compression force was monitored and presented as N/needle. To visually assess the morphologic behavior after mechanical tests, the morphology at selected displacement points was observed using an SEM (FEIE-SEM Quanta 200, Eindhoven, The Netherlands). The insertion of the MNs into the skin was assessed ex vivo using fresh skin taken from C57 mice. Each tested MN was inserted into the external side of the skin by applying a force of 1.6 N using an Mpatch mini applicator (Micropoint Technologies Pte Ltd., Singapore). Then, skin samples were fixated in 10% formalin for 24 h, soaked in OCT compound as a freezing medium for 2 h at room temperature, and transferred to −80 °C for 24 h. The frozen OCT skin samples were sliced into 10-µm-thick slices using a cryostat (Leica, Wetzlar, Germany). Slides were washed with double distilled water for 2 min (to remove remaining OCT) and stained with H&E. Finally, slides were covered with a mounting medium (xylene based) using a wood rod and coverslip. Insertions of the MNs were observed using an imaging reader equipped with an automated digital microscope (Cytation 5, Agilent, Santa Clara, CA, USA).

### 2.3. Release of Dex

The release rate of Dex from the MN across fresh porcine skin was determined using Franz diffusion cells (PermeGear, Hellertown, PA, USA). Skin samples were prepared by placing the skin tissue in a 70 °C water bath for 2 min (to allow separation of the skin membrane from the fat tissue) and carefully peeling the outer skin layer [30]. MNs were inserted into the outer side of the membrane and the diffusing cell was filled with PBS (pH = 7.4). The diffusion cell apparatus was kept at 37 °C with shaking at 200 rpm. Medium samples (3 mL) were taken at predetermined time points for analysis and replaced by fresh PBS. Each specimen was dried by lyophilization and rehydrated with 300 μL PBS in order to concentrate the samples. The released Dex was determined using high performance liquid chromatography (HPLC) with a C18 XBridge BEH column (4.6 mm × 150 mm, 130 Å, 3.5 μm) and a 2489 UV/vis detector (Waters, Milford, MA, USA) at 242 nm. The mobile phase contained acetonitrile and sodium acetate trihydrate at pH = 4.8 (60:40); the flow rate was 1 mL/min (n = 4). The drug release was calculated by comparison with a calibration curve. To observe the release process of active molecules from the fabricated MNs, rhodamine B solution (1 mg/mL, each array contained 0.11 mg) was loaded into MNs and placed on a porcine skin gelatin surface, and the release was observed and monitored using ImageJ software (V 1.8.0).

### 2.4. Cell Toxicity

Cytotoxicity was examined by exposing the MNs to NIH 3T3 fibroblast cell lines. Cells were grown in Dulbecco’s modified Eagle’s medium (DMEM) supplemented with 10% fetal bovine serum at 37 °C in a 95% air/5% carbon dioxide atmosphere at 95% relative humidity. MNs were initially dissolved in DMEM at increased concentrations. Cells were allowed to grow for 24 h, and the medium was replaced with the respective MN solutions. The cytotoxicity was assessed by an MTS assay using CellTiter 96 solution 24 h after exposure. The results are presented as averages and are expressed as a percentage ± stdv of the control. Four replicates were seeded for each of the tested MN, as well as for the control.

### 2.5. In Vivo Study

Animal experiments were approved by the Council for Animal Experiments, Israel Ministry of Health (animal ethics number: IL-100-07-22), in conformity with the guidelines of the Animal Welfare Law (published in 1994). Eighteen 7-week-old female BALB/c nude mice were purchased from Envigo (Jerusalem, Israel). All mice were maintained in sterilized cages in a 12 h light/12 h dark cycle with food and water provided ad libitum. The AD skin model [31] was induced by dropping 100 µL of 1% DNCB in acetone:olive oil (4:1) solution on the center of the back for 3 consecutive days. After 5 days (on day 8 after first sensitization), mice were randomly divided into three groups of six animals each: not-treated, empty-MNs, and Dex-MNs (N = 6). Skin sensitization was continued for all mice with 100 µL of 0.6% DNCB every other day. Treated groups (empty-MNs or Dex-MNs) were administrated daily for an additional 16 days with two MNs placed adjacent to each other. The severity of AD-like symptoms was assessed daily by a trained dermatologist (R. I.) and symptoms were scored according to the validated SCORAD assessment with an established index [32]: 0—None, 1—Slight, 2—Moderate, and 3—Severe. This evaluation method encompasses scoring of four clinical parameters that represent the level of inflammation in the skin: erythema, edema, dryness, and ulceration. Mice were sacrificed after 25 days, and spleens were harvested and measured using ImageJ. Skin samples were taken from the dorsal skin, fixated in 10% formalin, and stained with Toluidine Blue and H&E. Histological evaluations, including epidermal thickness and invasiveness of mast cells, were performed by a board-certified toxicological pathologist (A. N.). Histopathological changes were described and scored using a 5-point semi-quantitative grading (0–4), taking into consideration the severity of the changes. The score reflects the predominant degree of the specific lesion seen in the entire field seen in the histology section [33].

### 2.6. Statistics

Statistical evaluations of the compression and tension tests, drug release, cell viability, and in vivo results were performed using Prism 5 (GraphPad, La Jolla, CA, USA). Significant differences between groups were determined by a t test; *p* values < 0.01 and n = 4 for the in vitro characterization and n = 6 for the in vivo studies.

## 3. Results and Discussion

### 3.1. Formation and Characterization

Dex-loaded MNs were fabricated using the micromolding technique. Aqueous polymeric solutions (10 %wt) of various PVA/PVP ratios were mixed with Dex to a final concentration of 0.25%. A topical Dex preparation of 0.25% is commonly used worldwide in commercial creams and ointments (e.g., desoximetasone cream, USP and dexamethasone ointments, and USP from Taro, Haifa, Israel). This concentration was found to be effective for treating AD with minimal to no adverse effects [34]. The Dex solution was casted on a PDMS mold, spun at 4000 rpm, and dried for 24 h at room temperature (Figure 1A). Control MNs of the same composition but without Dex were fabricated using the same technique. Empty MNs showed a well-defined structure with pyramidal morphology (Figure 1B,C and Appendix A). Dex did not influence the formation of arrays with the loaded drug, as shown by SEM and by the naked eye (Figure 1D,E).

### 3.2. Mechanical Properties

In order to pierce the skin, MNs should possess adequate mechanical properties; they should be stiff in order to penetrate through the upper skin but also have some degree of elasticity in order to achieve intimate contact with the contours of the upper skin [35]. The effect of MN composition on the strength was measured by fixing the tested MN on the lower crimp and pressing the upper metal crimp on it. All MNs demonstrated tip deformation with no fracturing or breaking points, as was evident from the continuous force-travel curves (Figure 2A) [36,37]. MNs made from pure PVP exhibited the highest compression stiffness, with an average of 1 N/MN. The excellent mechanical properties of the high molecular weight PVP have been attributed to its small pores and narrow pore distribution in the solid matrix [38]. A previous study that examined the suitability of high (360 kDa) and low (10 kDa) molecular weight PVP for MN fabrication revealed that the pyrrolidone rings contribute to intramolecular rigidity, and that the 360 kDa PVP provides the highest compression strength [39]. The addition of the low molecular weight PVA to the formula decreased the compression force of the needles. For example, increasing the PVA content from 50% to 66.6% or 100% decreased the compression force from 0.68 to 0.56 and 0.38 N/MN, respectively. We note that MNs made from pure PVP were brittle compared with those containing PVA. This behavior was demonstrated in tension tests (Appendix A) and by the fact that about 25% of the MNs broke or could not be removed completely from the PDMS mold. Consequently, MNs with a PVA/PVP ratio of 1:2 were chosen for further analysis based on their excellent mechanical properties of a tough structure allowing easy and complete penetration whilst holding the required degree of elasticity. In fact, some degree of elasticity also enables intimate coupling to the skin, and therefore significantly enhances adhesion and stability of oral patches [40]. This is in particularly important in AD, where the skin tends to be irregular, dry, thickened, and occasionally lichenified [41].

In order to evaluate the possible effect of 0.25% Dex on the compression strength of the needles, PVA/PVP (1:2) MNs containing 0.25% Dex were fabricated (Figure 2B). The presence of the drug had only a negligible effect on the compression strength and the needles exhibited no breaking points or bending, as is evident from the continuity of the plot. This was also confirmed by SEM images collected during the experiment, which showed tip deformation with no evidence of fracturing or breaking points (Figure 2C). Finally, histological examinations demonstrated the successful insertion of the Dex-MN into mice skin (Figure 2D).

### 3.3. Release of Dex from MNs

Dex release into the PBS was measured using Franz diffusion cells using porcine skin followed by HPLC analyses (Figure 3A,B). Porcine skin is frequently used as an ex vivo model owing to its similarity compared to human skin in terms of fat deposits, follicles, and density [42]. From a practical point of view, the surface of the porcine skin is easily accessible and allows multi-sample extractions. About 70% of the drug was released within 1 h, followed by a more sustained release pattern for up to 4 h. A total of 94% of the drug was released after 5 h (Figure 3B). The relatively fast release of Dex from the MNs can be explained, in part, by the solubility of the hydrophilic 1:2 PVA/PVP composition [43]. The hygroscopic nature of both polymers also contributes to the fast release profile. After the MN is inserted into the skin, it absorbs some moisture, which results in swelling (Appendix A), dissolution, and release of the drug [28,44]. The permeability rate of the outer skin layer may be influenced by the thermal treatment. Therefore, we conducted a second study where the release pattern of rhodamine B from MNs placed in porcine skin gelatin was evaluated. The release pattern, as can be seen in Figure 3C,D, mirrored that seen ex vivo; a fast release was observed in the first 1 h, followed by a more moderate release rate in the following 2 h, as was determined by an increased rhodamine B releasing area. Such immediate release upon MN insertion was found to be clinically beneficial when targeted drug delivery is required [45]. This is also the case when treating AD, in which skin lesion symptoms can be relieved instantly [46,47]. Overall, the in vitro release experiment implies that MNs exhibit a satisfactory targeted burst behavior as well as total release pattern.

### 3.4. Cell Toxicity

The cytotoxicity of empty 1:2 PVA/PVP MNs was evaluated on skin NIH 3T3 fibroblast cells using the MTS method (Figure 4). MNs (10 mg) were dissolved in DMEM at concentrations between 3.125 mg/mL and 100 mg/mL. Cells (100,000 cells/mL) were exposed to polymer solutions in a 96-well plate and an MTS assay was performed. Figure 4A shows the ratios of viable cells relative to control cultures. Cells exposed to MN solutions at concentrations below 12.5 mg/mL showed no cytotoxicity, whereas above 25 mg/mL, cells displayed a gradual decrease in viability (Figure 4A). The differences between the tested compositions were also discernible in terms of actual appearance (Figure 4B). The observed toxicity of polymer solutions at high concentrations is attributed to the low availability of water to cells due to the high viscosity of the polymers in DMEM solution [19]. Notably, the concentrations assessed here are an order of magnitude higher than the relevant clinical concentration. For example, in recently published research studying the cytotoxicity of PVA/PVP MNs of various ratios, 1.52 mg/mL was chosen as the highest concentration assessed. Both PVA and PVP have excellent profiles in terms of their low toxicity and biocompatibility. Furthermore, according to previous reports, PVA and PVP dissolvable MNs showed a low cytotoxicity and no appreciable toxicity in mice after daily administration for 160 days [39,48,49]. These polymers are widely used in the fabrication of advanced biomedical systems and devices, as well as in cosmetics and wound dressings [50].

### 3.5. In Vivo Study

The clinical efficacy of the Dex-MNs was evaluated using an AD mice model (Figure 5A). An AD skin model was induced by the application of 1% DNCB solution for 3 consecutive days followed by a repeated challenge every other day. The allergen DNCB is internalized by local antigen-presenting cells, causing AD-like symptoms, including eczema, erythema, scaling, and hemorrhaging in the skin and increased infiltration of mast cells [51], as was observed in all AD-induced mice (Figure 5B). Groups receiving MNs (empty or with Dex) were administered daily with two arrays of MNs placed side by side in the center of the lesion (Appendix A). Both kinds of MNs immediately attached to the dorsal area upon administration, remaining there for about 3 h until fully dissolved. The non-treated and the empty-MNs groups presented severe erythema, slight-to-localized edema, and moderate epidermal ulcers (Figure 5B). Notably, the non-treated mice had severe dryness of the epidermis with flaking, while the empty-MNs group presented an improved condition, with only slight-to-moderate epidermal dryness (Figure 5D). The enhanced moisture-retention capacity of the mice treated with empty MNs can be attributed to the hygroscopic nature of PVA and PVP [52,53]. Several hygroscopic polymers, including PVA and PVP, have demonstrated the ability to absorb and retain moisture from the environment thanks to their hydrophilic structure and the presence of repeating hydroxyl and/or pyrrolidone units [54]. Moisturizing treatments applied directly to the skin were found to provide an occlusive barrier, thus protecting the skin from irritants. Increasing the humidity of the skin by formulated emollient products has been associated with antimicrobial, anti-itch, and anti-inflammatory actions [55]. The Dex-MNs group showed better clinical results compared with the other groups. Spleen enlargement, a common feature of chronic inflammation [56], was significantly suppressed in the Dex-MNs group, with spleens measuring around 40 mg and 15 mm on average, compared with around 90 mg and 20 mm, respectively, for the control groups (Figure 5C,E,F). Histological observations mirrored the clinical evaluation. The epidermal thickness of the Dex-MN group was found to be around 20 μm, compared with over 75 μm for the empty-MN and non-treated groups (Figure 5G). While neutrophil and eosinophil infiltration were not detected among the Dex-MN group, the empty-MN and non-treated groups had relative scores of 1.5 and 0.8, respectively (Figure 5H). Overall, our data suggest that Dex-MNs are capable of reducing both local and systemic physical inflammatory reactions elicited by DNCB.

## 4. Conclusions

A series of MNs loaded with Dex were fabricated and evaluated as a potential treatment for AD. PVA/PVP at a 1:2 ratio was chosen as the matrix after presenting the best mechanical and manufacturing properties. MNs demonstrated potential as drug carriers to the skin, biocompatibility with cell line cultures, and effectiveness in mice models. The strategy developed here provides a convenient means to overcome some of the shortcomings of traditional AD treatments. The introduction of a novel, reliable, cheap, and safe apparatus, the administration of which does not require trained medical staff, is expected to create lucrative opportunities for the treatment of AD and other cutaneous diseases and conditions.

## Figures and Tables

**Figure 1 pharmaceutics-15-01109-f001:**
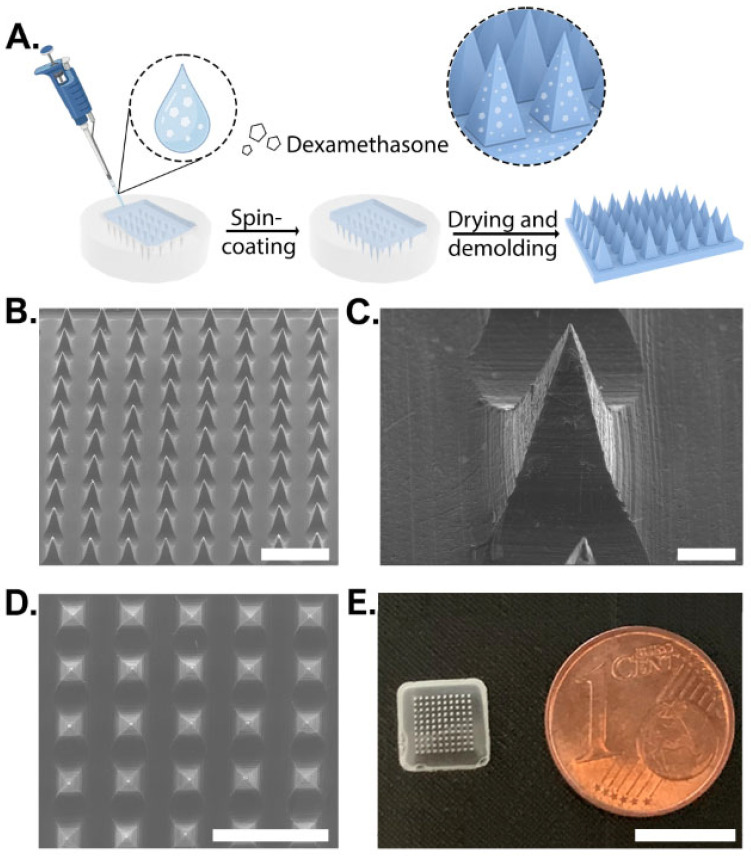
Schematic illustration of dexamethasone-loaded microneedle fabrication process and their morphology. (**A**) Manufacturing process using PDMS molding. (**B**) SEM image of microneedles containing dexamethasone. Scale bar: 1 mm. (**C**) SEM image of a well-structured MN. Scale bar: 100 µm. (**D**) Top view of SEM image. Scale bar: 1 mm. (**E**) Visual images of an MN. Scale bar: 8 mm.

**Figure 2 pharmaceutics-15-01109-f002:**
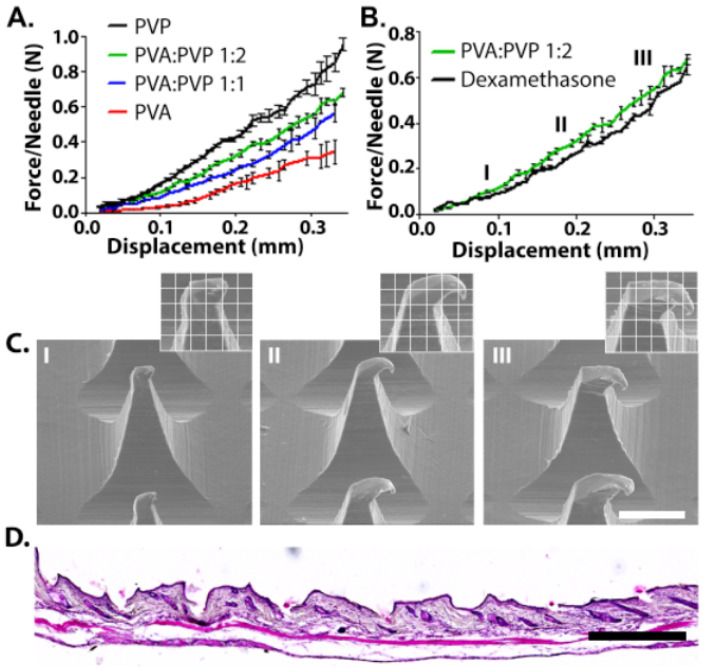
(**A**) Compression behavior of microneedles made from different ratios of PVA/PVP. (**B**) 0.25% Dex in 1:2 PVA/PVP MNs compared with empty MNs. (**C**) Representative SEM images of Dex-MNs after compression at selected points (I,II,III). Scale bar: 200 µm. (**D**) Histological examinations of mice skin after MN insertion. Scale bar: 500 µm.

**Figure 3 pharmaceutics-15-01109-f003:**
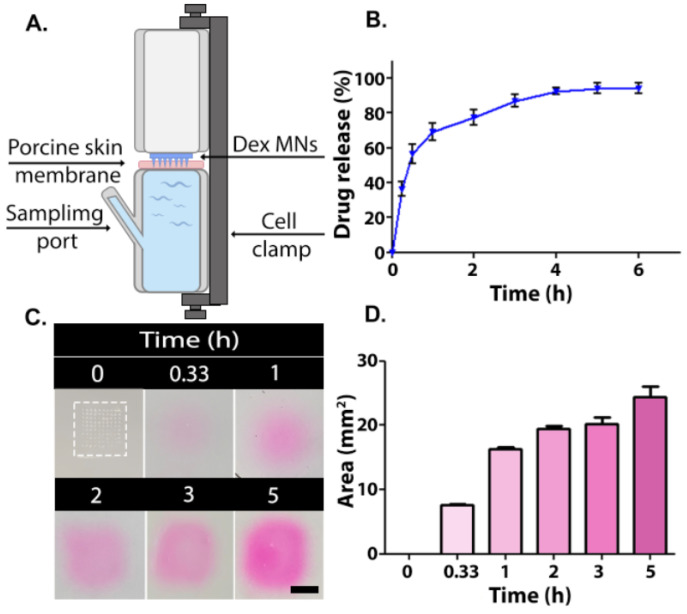
(**A**) Schematic illustration of a Franz diffusion cell. Release of active ingredient from MNs: (**B**) In vitro Dex release profile from 1:2 PVA/PVP microneedles (n = 4). (**C**) Gel images of rhodamine B release in pig skin gelatin (Microneedle is enclosed by dashed box). Scale bar: 50 mm. (**D**) Rhodamine B diffusion area measured by ImageJ.

**Figure 4 pharmaceutics-15-01109-f004:**
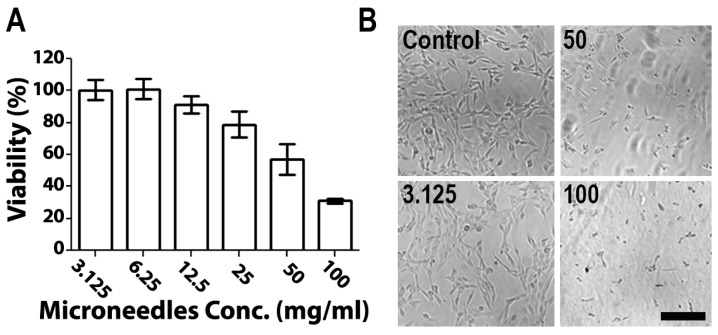
(**A**) NIH 3T3 cell viability compared with unexposed cells and (**B**) microscopic images of cells exposed to increased concentrations of microneedles. Scale bar: 200 µm.

**Figure 5 pharmaceutics-15-01109-f005:**
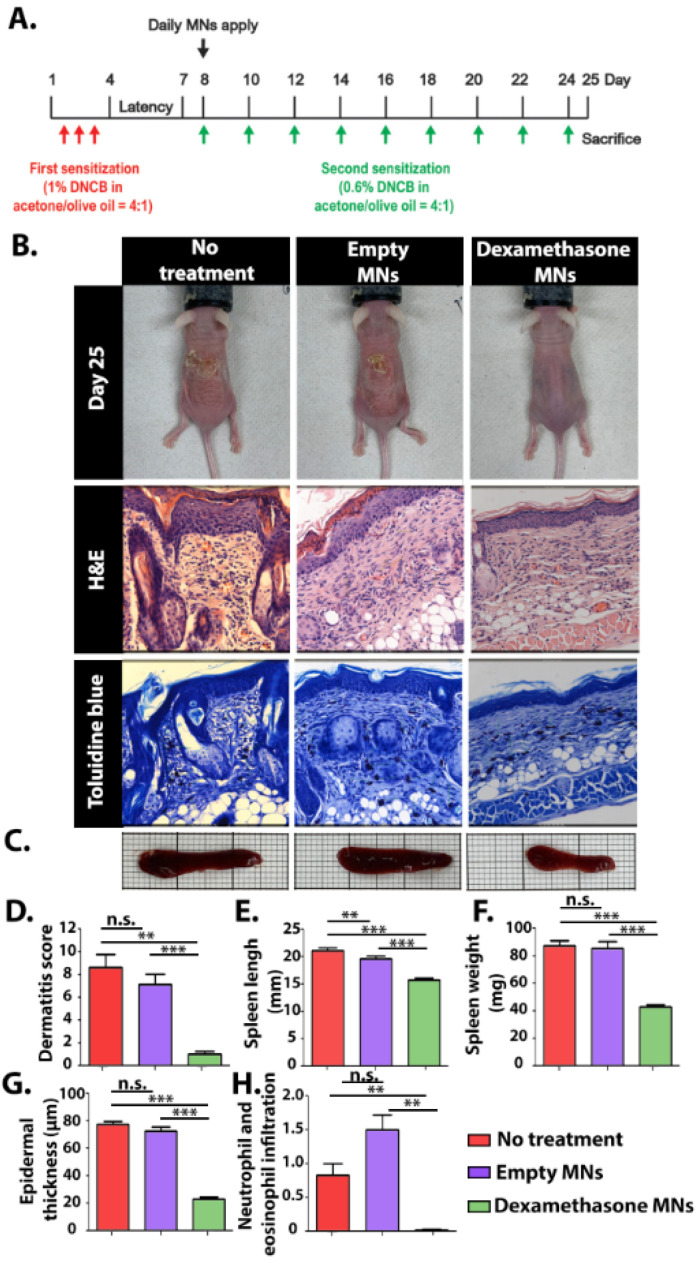
(**A**) Schematic representation of the experimental procedure. (**B**) Representative photos of the experimental mice after 25 days and histological images (magnification, ×400). (**C**) Representative images of the experimental spleens. In vivo evaluation at day 25: (**D**) dermatitis score, (**E**) spleen length, (**F**) spleen weight, (**G**) epidermal thickness, and (**H**) neutrophil and eosinophil infiltration. Data are means and SD of n = 6, ** *p* < 0.005, *** *p* < 0.0001, n.s.—non significant.

## Data Availability

The data presented in this study are available in this article and Appendix A.

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
