# Peer review of "Design and Evaluation of Dissolvable Microneedles for Treating Atopic Dermatitis"

_pharmaceutics, 2023, doi:10.3390/pharmaceutics15041109_

Round 1
Reviewer 1 Report
I would like to appreciate the authors for contribution on "Design and Evaluation of Dissolvable Microneedles for Treating Atopic Dermatitis". The work presented is interesting and scientifically sound. However, there are some minor modifications required to improve the quality of the manuscript.
Abstract: Can be improved and constructive: Aim of the study, Methods, results and conclusions
Introduction: Authors mentioned about dissolving microneedles, which are represented in the latest publications, Refer below:
Microneedle based natural polysaccharide for drug delivery systems: Progress and challenges: Pharmaceuticals 2022, 15(2), 190; https://doi.org/10.3390/ph15020190
"Multifunctional low temperature-cured PVA/PVP/citric acid-based hydrogel forming microarray patches: Physicochemical characteristics and hydrophilic drug interaction" https://doi.org/10.1016/j.eurpolymj.2023.111836"Biphasic burst and sustained transdermal delivery in vivo using an AI-optimized 3D-printed MN patch" https://doi.org/10.1016/j.ijpharm.2023.122647
Line 33: Replace Atopic Dermatitis with AD
I would request the authors to mention the animal protocol number.
What is the reason behind using 0.25% of the drug loaded into microneedles?
Authors also mentioned prior to animal studies processed the porcine skin at 70-degree C. what is the rationale behind this?
In this case the permeability is easy because of thermal treatment.
Ideally the release rates should have been compared to various drug concentrations or polymer concentrations. Please justify this?
In this study, the fabricated microneedles were optimally selected. How did the authors come to the conclusion?
Author Response
We thank the reviewer for this favorable review.
Question (1): Abstract: Can be improved and constructive: Aim of the study, Methods, results and conclusions
Answer: We have addressed the excellent request raised by the reviewer and restructured the abstract without deviating from journal’s format.
Question (2): Introduction: Authors mentioned about dissolving microneedles, which are represented in the latest publications, Refer below:
Microneedle based natural polysaccharide for drug delivery systems: Progress and challenges: Pharmaceuticals 2022, 15(2), 190; https://doi.org/10.3390/ph15020190
"Multifunctional low temperature-cured PVA/PVP/citric acid-based hydrogel forming microarray patches: Physicochemical characteristics and hydrophilic drug interaction" https://doi.org/10.1016/j.eurpolymj.2023.111836
"Biphasic burst and sustained transdermal delivery in vivo using an AI-optimized 3D-printed MN patch" https://doi.org/10.1016/j.ijpharm.2023.122647
Answer: We mentioned the reports as was suggested by the reviewer. Please see on page 2.
"…Since dissolvable MNs can painlessly bypass the skin's epidermis, dissolved at the treated area and release drugs directly to the inner skin layers [1-3], they are often used as common drug delivery systems [4]."
Question (3): Line 33: Replace Atopic Dermatitis with AD
Answer: We have replaced Atopic Dermatitis with AD.
Question (4): I would request the authors to mention the animal protocol number.
Answer: In response to the reviewer’s comment, we have added the animal protocol number. Please see on page 5.
“Animals experiments were approved by the Council for Animal Experiments, Israel Ministry of Health (Animal ethics number: IL-100-07-22), in conformity with the guidelines of the Animal Welfare Law (published in 1994).”
Question (5): What is the reason behind using 0.25% of the drug loaded into microneedles?
Answer: We have addressed the excellent question raised by the reviewer. Please see on page 5.
“Topical Dex preparation of 0.25% is commonly used worldwide in commercial creams and ointments (e.g. Desoximetasone Cream, USP and Dexamethasone ointments, USP from Taro, Israel). This concentration was found to be effective for treating AD with minimal to no adverse effects [34]. “
Question (6): Authors also mentioned prior to animal studies processed the porcine skin at 70-degree C. what is the rationale behind this? In this case the permeability is easy because of thermal treatment.
Answer: The purpose of the preliminary treatment is to allow separation of the skin membrane from the fats tissue. After a heat separation process, the epidermis and partial dermis (approximately 800 μm thick) can be peeled from the cadaver skin using tweezers. This protocol was well studied and found solid [6]. Nevertheless, we agree with the reviewer that permeability may be compromised due to the heat treatment and therefore have conducted a second in vitro study (Please see Figure 3C and D) which validated our results. We made mention these important comments in the text.
Page 4:
“Skin samples were prepared by placing the skin tissue in a 70 °C water bath for 2 min (to allow separation of the skin membrane from the fats tissue) and carefully peeling the outer skin layer [6]”.
And Page 8:
“The permeability rate of the outer skin layer may be influenced by the thermal treatment. Therefore, we have conducted a second study where the release pattern of rhodamine B from MNs placed in porcine skin gelatin was evaluated. Release pattern, as can be seen in Figures 3C and D mirrored that seen ex vivo: fast release in the first 1 h followed by a more moderate release rate in the following 2 h as was calculated by increased in rhodamine B releasing area”.
Question (7): Ideally the release rates should have been compared to various drug concentrations or polymer concentrations. Please justify this?
Answer: We agree with the reviewer that ideally the release rates should have been compared to various drug concentrations or polymer concentrations. However, since both PVA, PVP and dexamethasone-21-phosphate disodium salt are water soluble a, increasing the concentration of either one of them should not have a significant effect on the release pattern.
Question (8): In this study, the fabricated microneedles were optimally selected. How did the authors come to the conclusion?
Answer: The reviewer raises an excellent question. To address the reviewer comment we added an explanation to our selection in the text. Please see on page 7.
“Consequently, MNs with a PVA/PVP ratio of 1:2 were chosen for further analysis based on their excellent mechanical properties: tough structure allowing easy and complete penetration whilst holding the required degree of elasticity. In fact, some degree of elaticity also enables intimate coupling to the skin, and therefore significantly enhances adhesion and stability of oral patches [7]. This is in particularly important in AD, where the skin tend to be irregular, dry, thickened and occasionally lichenified [8]”.
Reviewer 2 Report
This work developed a new delivery system for AD treatment based on dissolvable microneedles containing dexamethasone incorporated in a dissolvable polyvinyl alcohol/polyvi-nylpyrrolidone matrix. This work is interesting and the results were clearly presented. However, there are still some significant points of criticism.
1 As mentioned in the profile, atopic dermatitis can lead to severe inflammation, but subsequent research on the level of inflammation in vivo and in vitro was limited.
2 At the molecular level, the expression levels of proteins related to healing should be explored , such as skin polymerin and sealing protein
3 There is a lack of biosafety studies at the animal level after prolonged use of microneedles.
4 The reasons and advantages of using fresh porcine skin as a measure of release rate of Dex from the MN should be explained.
Author Response
We thank the reviewer for this overall favorable review.
Question (1): As mentioned in the profile, atopic dermatitis can lead to severe inflammation, but subsequent research on the level of inflammation in vivo and in vitro was limited. At the molecular level, the expression levels of proteins related to healing should be explored , such as skin polymerin and sealing protein
Answer: We thank the reviewer for the important comment. In our study, we used a well-characterized and reliable animal model for atopic dermatitis, utilizing the best accepted evaluation parameters to test the level of inflammation. Namely, we have evaluated the clinical effect of the microneedles, using the validated SCORAD assessment tool. This tool encompasses scoring of four clinical parameters that represent the level of inflammation in the skin - erythema, edema, dryness, and ulceration. These parameters are also part of the clinical evaluation of atopic dermatitis in humans, and therefore serve as an excellent model for assessing the severity of inflammation. More details on this scoring system have been added to the manuscript text.
Furthermore, we have also used histopathology to evaluate the level of inflammation - assessing the epidermal thickness and neutrophil infiltration - both are well-characterized parameters for inflammation and have been used in the past to assess AD-like inflammation in mice. The inflammatory burden was also tested by measuring spleen weight, as a marker for systemic inflammation.
We believe that all these tests provide a reliable assessment of the inflammatory parameters that are part of the AD process. The expression levels of proteins related to healing are also very important parameter, however, they are beyond the scope of this single article.
Please see on page 5.
“Severity of AD-like signs were assessed daily by a trained dermatologist (R. I.) and symptoms were scored according to validated SCORAD assessment with an established index [9]: 0- None, 1-Slight, 2-Moderate, 3-Severe. This evaluation method encompasses scoring of four clinical parameters that represent the level of inflammation in the skin - erythema, edema, dryness, and ulceration.”
Question (2): There is a lack of biosafety studies at the animal level after prolonged use of microneedles.
Answer: In this manuscript we describe a new delivery system from fabrication to animal studies. Biosafety studies at the animal level are extremely important, but are beyond the scope of this (and perhaps any) single article. We note that according to previous reports, PVA and PVP dissolvable MNs showed low cytotoxicity and no appreciable toxicity in mice after daily administration for 160 days [10-12].
Please see on page 9-10:
“Both PVA and PVP have excellent profiles in terms of their low toxicity and biocompatibility. Furthermore, according to previous reports, PVA and PVP dissolvable MNs showed low cytotoxicity and no appreciable toxicity in mice after daily administration for 160 days [10-12].“
Question (3): The reasons and advantages of using fresh porcine skin as a measure of release rate of Dex from the MN should be explained.
Answer: We have addressed the excellent comment raised by the reviewer. Please see on page 8.
“Porcine skin is frequently used as ex-vivo model owing to its anatomical similarity compared to human skin in terms of fat deposits, follicles, and density [13]. From practical point of view, the surface of the porcine skin is easily accessible and allows multi-sample extractions.”
Author Response
We thank the reviewer for this favorable review.
To be considered for publication in Pharmaceuticals, the author needs to make revisions of this manuscript:
Question (1): What is importance of PVA/PVP in dissolving microneedles. Dissolving microneedles were made of various kinds of soluble polymer. Why PVA/PVP was chosen and the meaning of selection of PVA/PVP as a dissolving microneedle material?
Answer: In this manuscript, PVA/PVP was selected as the carrier for the polymer’s water solubility, hygroscopic properties, ability to easily adhere to the skin, and safety for cells and tissues [14-16]. We added the following data to the revised manuscript, Please see on Page 2:
…”Both PVA and PVP are considered ideal for fabricating fibers, films, and MNs due to their semi crystalline structure and excellent physical properties [14,17,18]. In addition, the hygroscopic properties of the polymers are providing moisturizing addition when used in dermal application [19]. MNs fabricated from PVA only might have insufficient mechanical properties to penetrate the stratum corneum. In order to increase the mechanical strength of the MNs, addition of another polymer (e.g. PVP) to the MNs composition is often efficient. MNs made from a mixture of PVA and PVP present improved mechanical properties since a hydrogen bond can be formed between the carbonyl group of PVP and the hydroxyl group of PVA [20].
Question (2): More experiments should be performed to support the manuscript, such as pharmacokinetics or pharmacodynamics studies.
Answer: In this manuscript we describe a new delivery system from fabrication to animal studies. The reviewer is correct that pharmacokinetics and pharmacodynamics studies are extremely important. However, respectfully, we feel that these studies are beyond the scope of this (and perhaps any) single article.
Question (3): The author needs to focus on the scale uniformity in the figures.
Answer: We have corrected the scale uniformity in the figures.
Question (4): In Fig.2C, the author appears to depict various degrees of needle tip curvature. We recommend the author to add a few auxiliary lines to make it more understandable.
Answer: We have added a few auxiliary lines and explanation in the figure legends.
Question (5): Some parts of the fonts in figures are too small.
Answer: We have corrected the fonts in figures.
Question (6): On line 333, the number should be written as 3.5 rather than 3.7.
Answer: The typos mentioned by the reviewer has been corrected.
Question (7): Written English should be improved.
Answer: An English editor has revised the entire revised text
Round 2
Reviewer 3 Report
The authors have properly responded all my questions. There are no further questions in the current manuscript, and could be consider to publication.Author Response
We thank the reviewer for this favorable review.
To be considered for publication in Pharmaceuticals, the author needs to make revisions of this manuscript:
Question (1): What is importance of PVA/PVP in dissolving microneedles. Dissolving microneedles were made of various kinds of soluble polymer. Why PVA/PVP was chosen and the meaning of selection of PVA/PVP as a dissolving microneedle material?
Answer: In this manuscript, PVA/PVP was selected as the carrier for the polymer’s water solubility, hygroscopic properties, ability to easily adhere to the skin, and safety for cells and tissues [14-16]. We added the following data to the revised manuscript, Please see on Page 2:
…”Both PVA and PVP are considered ideal for fabricating fibers, films, and MNs due to their semi crystalline structure and excellent physical properties [14,17,18]. In addition, the hygroscopic properties of the polymers are providing moisturizing addition when used in dermal application [19]. MNs fabricated from PVA only might have insufficient mechanical properties to penetrate the stratum corneum. In order to increase the mechanical strength of the MNs, addition of another polymer (e.g. PVP) to the MNs composition is often efficient. MNs made from a mixture of PVA and PVP present improved mechanical properties since a hydrogen bond can be formed between the carbonyl group of PVP and the hydroxyl group of PVA [20].
Question (2): More experiments should be performed to support the manuscript, such as pharmacokinetics or pharmacodynamics studies.
Answer: In this manuscript we describe a new delivery system from fabrication to animal studies. The reviewer is correct that pharmacokinetics and pharmacodynamics studies are extremely important. However, respectfully, we feel that these studies are beyond the scope of this (and perhaps any) single article.
Question (3): The author needs to focus on the scale uniformity in the figures.
Answer: We have corrected the scale uniformity in the figures.
Question (4): In Fig.2C, the author appears to depict various degrees of needle tip curvature. We recommend the author to add a few auxiliary lines to make it more understandable.
Answer: We have added a few auxiliary lines and explanation in the figure legends.
Question (5): Some parts of the fonts in figures are too small.
Answer: We have corrected the fonts in figures.
Question (6): On line 333, the number should be written as 3.5 rather than 3.7.
Answer: The typos mentioned by the reviewer has been corrected.
Question (7): Written English should be improved.
Answer: An English editor has revised the entire revised text